# Keyframe Insertion: Enabling Low-Latency Random Access and Packet Loss Repair

**Glenn Van Wallendael** [1,*] , **Hannes Mareen** [1] , **Johan Vounckx** [2] **and Peter Lambert** [1]

1    Department of Electronics and Information Systems, Ghent University—imec—IDLab, Technologiepark-Zwijnaarde 122, 9052 Gent, Belgium; hannes.mareen@ugent.be (H.M.); peter.lambert@ugent.be (P.L.)
2    THEO Technologies, Philipssite 5, bus1, 3001 Leuven, Belgium; johan.vounckx@theoplayer.com
*    Correspondence: glenn.vanwallendael@ugent.be

**Abstract:** From a video coding perspective, there are two challenges when performing live video distribution over error-prone networks, such as wireless networks: random access and packet loss repair. There is a scarceness of solutions that do not impact steady-state usage and users with reliable connections. The proposed solution minimizes this impact by complementing a compression-efficient video stream with a companion stream solely consisting of keyframes. Although the core idea is not new, this paper is the first work to provide restrictions and modifications necessary to make this idea work using the High-Efficiency Video Coding (H.265/HEVC) compression standard. Additionally, through thorough quantification, insight is provided on how to provide low-latency fast channel switching capabilities and error recovery at low quality impact, i.e., less than 0.94 average Video Multimethod Assessment Fusion (VMAF) score decrease. Finally, worst-case drift artifacts are described and visualized such that the reader gets an overall picture of using the keyframe insertion technique.

**Keywords:** H.265/HEVC; keyframe insertion; random access; error recovery; packet loss; channel switching

## 1. Introduction

Low-latency video distribution is a challenging domain because requiring a minimal buffer in all the components of the distribution chain drastically restricts the technological solutions on offer. Additionally, in such a distribution scenario, there are numerous end users with different device and connectivity characteristics. With respect to devices and connections, it is of the uttermost importance to not let the low-performing devices and connections badly influence the service of their high-performing counterparts. Especially when such distribution takes place over wireless unreliable networks, a large diversity in quality of service is unavoidable, even for the same end user who can be physically moving over time.

When further looking at end user behavior, the possibility for random access should be taken into account as well [1]. Random access, channel switching, or zapping is the process of starting to decode a previously unseen video stream. Again, the instantaneous and temporary nature of such random-access events should only minimally influence steady-state viewing performance. It is considered good practice to keep zapping time below 0.43 s [2]. In a different study, they observed that at zapping times of one second, more than half the participants notice the delay, and starting at 4 s, most participants start to get annoyed [3].

The fundamental problem can be found in the combination of these low-latency video distribution requirements and the frame prediction paradigm of video compression. Video compression mainly earns its compression efficiency gains from using inter-frame dependencies. By predicting blocks of pixels from one frame to the other, dependencies

between successive frames emerge. Although these dependencies provide enormous compression efficiency gains, they make the video stream susceptible for packet loss and random-access restrictions.

In general, when distributing video over a network, there are two reasons to break the inter-frame dependencies inside the video stream: for random access and for packet loss repair. Breaking this dependency is performed by introducing costly intra-predicted frames (I-frames) or keyframes. In our experiments in Section 4.2, we show that, depending on the encoder configuration, these keyframes can be 7 to 30 times larger in size compared to predicted frames. Therefore, they should be used scarcely.

In this paper, we describe and thoroughly evaluate a technique we named keyframe insertion. The main idea is to separate compression efficiency from random access and error resilience such that individual channel switching or low performance does not influence the larger group of high-performing end users. This separation is reflected in the generation of two video streams: a normal stream (NS) and a companion stream (CS). The normal stream consists of an efficiently compressed video stream including a minimal amount of keyframes. As keyframes are needed for random access and error recovery, this normal stream is accompanied by a companion stream consisting of keyframes only. Recovering an error or starting to watch a video stream translates to retrieving one single keyframe from the companion stream followed by decoding of the normal stream. It is important to observe that by splicing an intra-predicted frame in place of an inter-predicted frame causes a decoded pixel mismatch resulting in drift artifacts.

In summary, although previous works introduced the idea of inserting keyframes in video streams for channel switching purposes in Motion Picture Experts Group (MPEG) MPEG-2 [4] and briefly in Advanced Video Coding (H.264/AVC) [5], this is the first work:

- To apply the basic and mainly untested concepts to a recent compression standard, namely, H.265/HEVC, and to provide restrictions and modifications necessary to make the basic idea working under this recent compression standard.
- To thoroughly quantify the quality (VMAF, Structural Similarity (SSIM), and Peak Signal-to-Noise Ratio (PSNR)) and bitrate impact, such that realistic performance of the keyframe insertion technique can be anticipated.
- To quantify and analyze the impact of using the reconstructed normal stream to encode the companion stream rather than using the original source video as input.
- To visualize the specific artifacts caused by the decoding error drift after keyframe insertion. This is important because the traditional averaging of objective measures may mask the artifacts that this system could confront end users with.

In our opinion, the results from this work provide video distribution system architects with the necessary insight to consider and evaluate the proposed keyframe insertion technique.

This paper proceeds with a thorough investigation of state-of-the-art techniques used for random access and packet loss repair in Section 2. Details on the materials and methods of this work are provided in Section 3 followed by a comprehensive description of the results in Section 4. Finally, this work is completed with an example application in Section 5 and a discussion in Section 6.

## 2. State-Of-The-Art

Solutions to increase random access capabilities and packet loss recovery can be divided in three categories, depending on which element in the video distribution chain needs modification: client-based, network-based, and content-based. The proposed work is related to the content-based approaches described here.

### 2.1. Client-Based Methods

Purely on the client device of the end user, so without modifications to the content or the network, techniques for fast channel switching can be categorized as prefetching

techniques and playback modification techniques. With respect to packet loss mitigation, error concealment can be considered an alternative to the proposed technique.

A lot of prefetching techniques have been studied, which try to predict the channels where the user is going to change towards for Internet Protocol (IP)-based techniques [6] and for Digital Video Broadcasting (DVB) [7]. Some techniques prefetch a low-resolution version to support a preview mode [8]. Prefetching can also be performed by sequentially sampling video segments from a set of target channels that a client might switch to in the near future [9]. These video segments downloaded from the combination channel will be cached in a local buffer. When low-latency requirements are considered, prefetching would require a lot of simultaneous open streams resulting in a large bitrate and processing burden on the client device.

Alternatively, research shows that there is a possibility to vary the playback speed to enable smaller buffers and thus reduce latency [10]. This concept can be coupled to an early start of audio combined with slow playback of video [11]. Although useful, such techniques would only allow small changes to the buffering behavior of the end-to-end system and should be considered complementary to the proposed technique.

Whenever network-layer packets get lost, corrupted Network Abstraction Unit (NAL) packets should be discarded entirely. Numerous error concealment algorithms have been and are being developed for H.264/AVC [12,13]. For more recent standards like H.265/HEVC and Versatile Video Coding (H.266/VVC), there is little work on error concealment techniques [14]. The main problem with performing error concealment on these standards is the Temporal Motion Vector Prediction Tool (TMVP). Performing error concealment on pixels provides decent looking results, but errors in concealed motion vectors leads to large and annoying artifacts.

### 2.2. Network-Based Methods

Within the network, different types of processing can be performed to enable faster channel switching behavior. The type of processing can be separated into caching behavior and complex processing.

In-network caching allows the end user device to quickly receive past frames up to the last keyframe at the network speed. Although such caching eliminates the delay dependency to the buffer duration, network throughput becomes the bottleneck instead [15]. In an alternative work, a similar caching behavior is proposed, but the cached portions of the multicasted stream are delivered using unicast to the end-user device [16]. Additionally, this strategy is not able to solve any packet loss related problems.

When processing is allowed in the network, a solution can be found in a zapping accelerator server. Such a server generates several time-shifted replicas of the video stream from which one is sent until there is an I-frame in the regular channel [17]. Note that this paper considered generating these streams in the network rather than at the source, otherwise this work could be considered a content-based technique. In different works, transcoding at the cloud edge is proposed to transform the requested video to specific formats [18]. Transcoding videos at channel change events requires a lot of processing complexity and even with complexity reductions around 82% compared to reference encoders [19], such processing is excessively expensive. The only solution capable of limiting transcoding complexity with 99.2% uses coding information calculation (CIC) modules and residual encoder (RE) modules [20] such that only entropy decoding and encoding for each channel switching or packet loss event is required. In the proposed solution, we opt to keep the complexity in the network minimal.

### 2.3. Content-Based Methods

When considering the end-to-end system, the distributed video streams can be modified to accommodate channel switching and packet loss behavior. In the H.264/AVC standard, the concept of Switching Intra (SI) pictures and Switching Predicted (SP) pictures has been designed to facilitate random access and packet loss scenarios [21]. By modifying

regular P-pictures in the video stream to become SP-pictures, these SP-pictures can be interchanged with accompanying SI-pictures capable of reproducing exactly the same visual end result. Within the SI/SP picture strategy, to keep the SI-picture's size under control, an overhead is introduced in the SP-picture. This overhead results in reduced compression efficiency for the normal video stream, which is an aspect our proposed solution avoids. Additionally, SI/SP picture coding has not been widely adopted in H.264/AVC encoders and decoders and the compression tool has not been included in H.265/HEVC.

Different than the SI/SP picture strategy, all of the following techniques are based on the observation that it is more important to have a fast visual feedback on a channel change request, than the full quality immediately after tune-in [22]. To accommodate this, two video streams work together to provide random access and packet loss recovery. Similar to the proposed technique, one stream provides a high-quality representation with minimal keyframes (normal stream) and the other provides a large amount of keyframes (companion stream) to accommodate fast channel change and recovery [4,5]. Channel changes or loss recoveries then happen by borrowing a keyframe from the companion stream and inserting it in the normal stream. Similar principles have been applied for Gradual Decoder Refresh (GDR) or intra refresh coding such that the companion stream fills up missing I-slice information [23]. Additionally, others propose to include a picture-in-picture channel in the stream. This channel has a lower bit rate (and resolution) than the regular channel, but it is constructed with a small GOP size [24]. For the reader's reference, in previous works, the normal stream [5,25] has also been named the main stream [4]. The companion stream [26] is known in other works as the synchronization stream [4], the channel change stream [5], or the fast channel change stream [25].

Because both the normal stream and the companion stream contain the same pixel content, as an optimization, both streams can be compressed as a scalable representation. Most works propose to provide the low-quality companion stream as a base layer enhanced with the normal stream as enhancement layer [27]. During steady-state watching, these techniques use both the base and enhancement layers of the selected channel to achieve full quality. Fast channel switching and error recovery is then performed by decoding only the base layer containing numerous keyframes. The downside of this technique is the continuous presence of a low-efficiency base layer which causes an overhead associated with scalable coding. To solve this problem, single-loop scalable coding has been utilized in Scalable Video Coding (SVC) [26] and proposed for single-loop H.265/HEVC [28]. With single-loop scalable decoding, the normal stream can be configured as the base layer extended with companion stream keyframes in the enhancement layer. Steady-state viewing of the video stream will thus be efficiently provided as a single layer base layer video stream without the associated scalable overhead. Only during channel switching or error recovery, the enhancement layer would be necessary to provide keyframe functionality. Scalable coding did not see global adoption mainly due to implementation complexity, but also because of bitrate overhead.

## 3. Materials and Methods

The main idea is to generate a normal video stream for which random-access and packet-loss considerations are minimized (see Figure 1). This normal stream is accompanied by a companion stream designed to solve random access and packet loss situations. The benefit of this strategy is that such a separation in responsibility makes the design decisions about compression settings more straightforward.

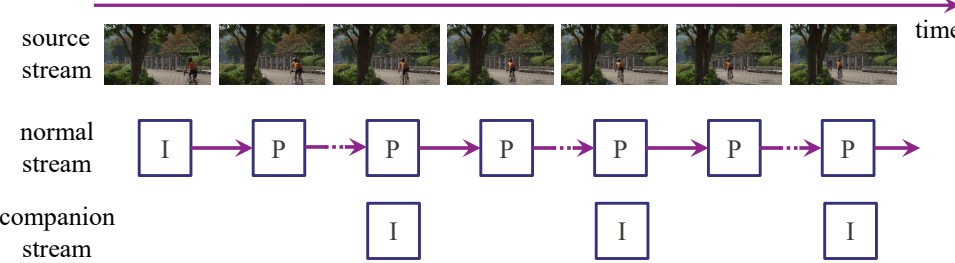

**Figure 1.** A source video is encoded as a low-latency normal video stream accompanied by a companion stream providing keyframes at specific intervals.

Within the normal stream, encoders will still introduce keyframes when content specific changes occur, such as high-motion events or scene changes. Bounding the interval between keyframes is necessary here to completely recover the normal stream from artifacts. On the other hand, limiting the number of keyframes is beneficial for compression efficiency resulting in an efficient normal video stream. Additionally, the companion stream is generated consisting solely of keyframes such as Instantaneous Decoder Refresh (IDR) or Continuous Random Access (CRA) frames. Both of these frametypes are keyframes as defined in H.265/HEVC. The frequency of these keyframes must be chosen corresponding to the required random-access latency, the packet loss recovery latency, and the bitrate overhead. The data to make such a trade-off are provided in Section 4.

Whenever a random-access event or packet-loss event occurs, the following procedure needs to be followed (see Figure 2). After a switch or loss event, let us denote the Picture Order Count (POC) of the first occurrence of a keyframe in the companion stream as $f$. At that moment, the keyframe at position $f$ will be used for starting the decoding process instead of the predicted frame at $POC = f$. After that, the regular decoding process can continue with decoding $f + 1$. With packet loss, this procedure assumes that the regular decoding process continues from the moment packets are lost (implying packet-loss drift errors) until the first keyframe in the normal stream or the companion stream, whichever comes first. Only then, at $POC = f$, the keyframe is decoded and regular decoding of the normal stream continues from $POC = f + 1$.

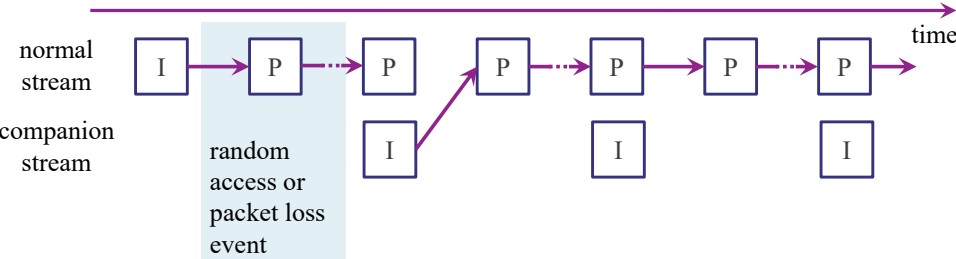

**Figure 2.** After a random access or packet loss event, a companion stream keyframe replaces the collocated predicted frame.

The normal stream will be encoded from the original source content (NS$_{src}$). For the encoding of companion streams, two possibilities for input selection will be considered and measured. First, the companion streams can be encoded from the source content similar to what happened with the normal stream (CS$_{src}$). Second, a closer match with the normal stream can be found by decoding the normal stream and encoding these reconstructed as a companion stream (CS$_{NS}$). This process is illustrated in Figure 3.

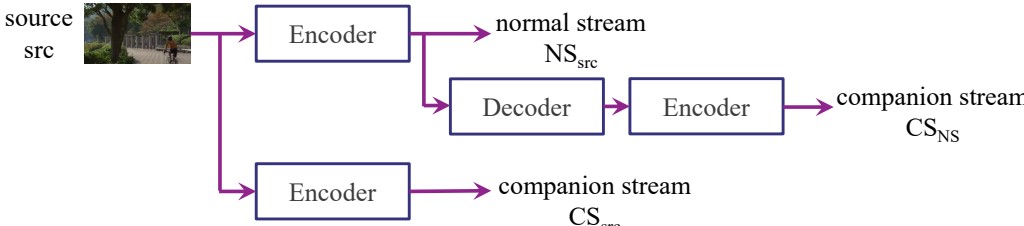

**Figure 3.** Companion stream CS$_{src}$ is encoded from the source video and CS$_{NS}$ is constructed from the decoded normal stream.

Although the basic idea of inserting keyframes has been mentioned in previous work with respect to MPEG-2 and H.264/AVC, it is important to consider the restrictions associated with the recent H.265/HEVC standard when performing the proposed operations.

- POC: When starting the decoding process at a companion stream keyframe, the POC of this keyframe should be identical to the POC of the replaced frame in the normal video stream. The synchronization of keeping these POCs identical is preferably performed during the encoding process, but can alternatively be performed right after the encoding process through an entropy rewrite step.
- Resolution: In contrast to H.266/VVC, H.265/HEVC cannot accommodate resolution changes within a Group of Pictures (GOP), such that the resolution of the normal stream and the companion stream should be identical.
- Sequence Parameter Set (SPS) and Picture Parameter Set (PPS): The parameter sets of both the normal and the companion stream should be identical or a retransmission of parameter sets should occur after the introduced keyframe from the companion stream.
- TMVP: Temporal motion vector prediction is a new tool introduced since H.265/HEVC and prevents the possibility to exchange predicted frames with keyframes. TMVP [29,30] enables borrowing motion vectors from reference frames additional to pixel information. Because of a lack of motion information present in keyframes, the frames succeeding the introduced companion keyframe will lack such motion information resulting in explicit and annoying blocking artifacts. For this reason, TMVP had to be disabled in the presented results. In general, whenever there is packet loss present during transmission, it is advised to disable TMVP because of the artifacts it produces. On the dataset used in this paper, enabling TMVP would lead to a Bjontegaard-Delta bitrate (BD-Rate) decrease of −2.57% (Std: 0.86%, Min: −3.92%, Max: −0.22%).

## 4. Results

This section presents and evaluates the experimental results concerning the proposed method based on keyframe insertion for H.265/HEVC. First, Section 4.1 describes the setup that was used to perform the experiments. Then, Sections 4.2 and 4.3 analyze the impact on the frame size and quality, respectively.

### 4.1. Experimental Setup

In order to evaluate the proposed method, 23 sequences with resolutions between $416{\times}240$ and $2560{\times}1600$ were used: *BlowingBubbles, BasketballPass, BQSquare, RaceHorses, BasketballDrill, BasketballDrillText, PartyScene, RaceHorses, BQMall, Johnny, FourPeople, KristenAndSara, SlideEditing, SlideShow, ChinaSpeed, BasketballDrive, BQTerrace, Cactus, Kimono, ParkScene, ParkJoy, Traffic, PeopleOnStreet* [31]. Each sequence contains between 150 and 600 frames and has a frame rate between 20 and 60 frames per second.

For encoding the sequences, the HEVC reference Model (HM) version 16.15 was used. For NS$_{src}$, a low-delay configuration was used in which the first frame is an intra-frame and all other frames are predicted frames that each take only a single preceding frame as reference. As discussed above, TMVP is turned off because inserting a keyframe disrupts

this coding tool. For $CS_{src}$ and $CS_{NS}$, the same configuration is used as for $NS_{src}$, yet only encoding intra-frames that do not depend on other frames.

In general, it is most straightforward to use the same QP for both the normal stream and companion stream. However, for completeness and additional flexibility, we also provide results of using different QPs. Therefore, each source sequence is compressed with four Quantization Parameter (QP) values (denoted as $QP_{NS}$ for $NS_{src}$, and $QP_{CS}$ for $CS_{src}$ and $CS_{NS}$): 22, 27, 32, and 37.

In the experiments, we inserted a keyframe at frame $f = 9$, which is the 10th frame and started decoding from there on as if a random access was performed or as if frame $f = 8$ had been lost. This is an arbitrary decision and could have been any other frame in the beginning of the sequence. Given the set of sequences we used, the 10th frame is taken to avoid duplicated frames and black frames present in the first frames of some sequences. These duplicate and black frames distorted the results heavily because their rate would be very small.

The keyframes from $CS_{src}$ and $CS_{NS}$ are used to replace an inter-frame of $NS_{src}$, while all other frames of $NS_{src}$ are left unchanged. These videos with the keyframe inserted (i.e., KI) are denoted as $KI_{src}$ and $KI_{NS}$, respectively. The effect of the keyframe insertion on the size and quality are evaluated in the next sections.

*4.2. Impact on Frame Size*

Table 1 shows the factor of frame size increase, i.e., the factor with which the size of the inserted keyframe is larger than the inter-frame of $NS_{src}$ that it replaces. The results in the table are averaged over all sequences, for both $KI_{src}$ and $KI_{NS}$, for all $QP_{NS}$, and for all $QP_{CS}$. When $QP_{NS} \neq QP_{CS}$, the results are shown in gray rather than black, as these may be less relevant in our application. Furthermore, note that the frame size increase depends on the used encoder configurations.

In absolute value, it can be observed that the inserted keyframes are between 2.0 and 149.4 times larger than the inter-frames that they replace. At equal QPs ($QP_{NS} = QP_{CS}$), the keyframes are between 7.4 and 29.2 times larger than their predicted counterparts. This gives a good indication about the bitrate burst that can be expected when such a companion stream keyframe is additionally requested over a network.

When comparing the frame size increases from $KI_{src}$ and $KI_{NS}$, one can observe that $KI_{NS}$ has a smaller size increase. That is because $KI_{NS}$ uses $NS_{src}$ as input, which has already been encoded and which thus contains less high-frequency noise. In other words, previously encoded content takes less bitrate to compress at equal QP settings.

Furthermore, a higher $QP_{NS}$ value (with an equal $QP_{CS}$ value) generally results in a larger size increase. This is because the ratio between intra- and inter-frames is larger in those cases, i.e., the overhead from intra-frames is larger for encodes with lower qualities. Last, when using a constant $QP_{NS}$ value, a higher $QP_{CS}$ value results in a lower bitrate increase, which is expected behavior.

**Table 1.** Average factor with which size of inserted keyframe increases compared to the inter-frame that it replaces.

| QP_CS | Average Factor of Frame Size Increase | | | | | | | |
| | KI_src | | | | KI_NS | | | |
| QP_NS = | 22 | 27 | 32 | 37 | 22 | 27 | 32 | 37 |
|---|---|---|---|---|---|---|---|---|
| 22 | 8.6 | 27.4 | 68.2 | 149.4 | 7.4 | 19.6 | 40.8 | 75.6 |
| 27 | 5.3 | 16.1 | 39.8 | 86.4 | 4.9 | 13.9 | 28.9 | 51.3 |
| 32 | 3.3 | 9.7 | 23.6 | 50.7 | 3.2 | 8.9 | 20.2 | 35.9 |
| 37 | 2.0 | 5.8 | 13.8 | 29.2 | 2.0 | 5.5 | 12.2 | 24.1 |

When looking at the frame size increase, note that keyframe insertion typically occurs infrequently. That is, a single (relatively large) keyframe is inserted only when packet loss

or channel switching occurs. In all other cases, no bitrate overhead occurs. Furthermore, note that alternative solutions either have a high latency or a higher bitrate overhead. First, one can wait for the next keyframe to resume playback, which may be a very long time (e.g., up to 600 frames). Not only does this have a high latency, it also wastes bandwidth potential during the waiting period. Second, an alternative solution for fast channel switching is to request the keyframe from the current segment at the new channel (from the normal stream), as well as all inter-frames up until the current frame. Then, these are all decoded and optionally displayed on-screen with a fast playback speed. This results in a high latency and requires more bandwidth than the proposed keyframe insertion (which requests only a single keyframe to start playback of the current frame at the new channel).

### 4.3. Impact on Quality

To measure the quality impact, the Peak Signal-to-Noise Ratio (PSNR) and Structural SIMilarity (SSIM) are commonly used [32]. Although these measures are well understood, they do not model the subjective quality accurately. More recently, the Video Multimethod Assessment Fusion (VMAF) was designed, which offers a better prediction of human quality perception [33]. The VMAF score is a number between 0 and 100, where 100 means that the two videos are subjectively indistinguishable. It has been claimed that a 6-point difference in VMAF score is just noticeable, i.e., it is a just-noticeable difference (JND) [34]. In this paper, the VMAF metric is mainly used, although some PSNR and SSIM scores are additionally given for completeness.

First, Section 4.3.1 analyzes the impact on the VMAF score. Then, Section 4.3.2 evaluates the number of frames before $KI_{NS}$ is of a better quality than $KI_{src}$. Then, the PSNR and SSIM results are discussed in Sections 4.3.3 and 4.3.4, respectively. Last, Section 4.3.5 discusses the worst-case VMAF scores.

#### 4.3.1. VMAF Decrease

First, the quality is evaluated in-depth on a single sequence. That is, Figure 4 shows the VMAF scores of the *ParkScene*-sequence, for $NS_{src}$, and for $KI_{src}$, and $KI_{NS}$. The videos are encoded with $QP_{NS} = QP_{CS} = 22$. As the keyframe is inserted at the 10th frame, the VMAF scores are equal in the first 9 frames.

At the 10th frame, the switch in $KI_{src}$ causes an increase in quality compared to $NS_{src}$. That is because the inserted keyframe is of a higher quality (and larger frame size) than the inter-frame that it replaced. However, this quality increase is quickly negated in the subsequent inter-frames; the quality decreases compared to the non-switched video. That is because these subsequent frames expect a different frame as reference. Even though the quality of the reference has increased, the predictions are wrong and hence errors are introduced. After a few seconds, the decrease in VMAF score compared to $NS_{src}$ stabilizes around 0.47. On average over the whole video (excluding the first 9 frames), the VMAF score decreases with 0.63.

In case of $KI_{NS}$, the keyframe insertion at the 10th frame immediately causes a decrease in quality compared to $NS_{src}$. In other words, the opposite effect of $KI_{src}$ is observed. That is because the inserted keyframe was encoded with $NS_{src}$ as input, and thus is inherently of a lower quality than the input itself. Although the inserted keyframe of $KI_{NS}$ initially decreases the quality of the video, the subsequent frames do not further decrease the quality significantly. Starting from the 14th frame, $KI_{NS}$ even has a better quality than $KI_{src}$. That is because $KI_{NS}$ is closer to the reference that the subsequent inter-frames expect for their predictions. Therefore, the errors that they introduce are smaller than those in $KI_{src}$. After a few seconds, the difference in VMAF score compared to $NS_{src}$ stabilizes to approximately 0.36. On average over the whole video (excluding the first 9 frames), the VMAF score decreases with 0.45. In other words, using the encoded $NS_{src}$ video as input for the inserted keyframe results in a smaller average quality decrease than using the source as input.

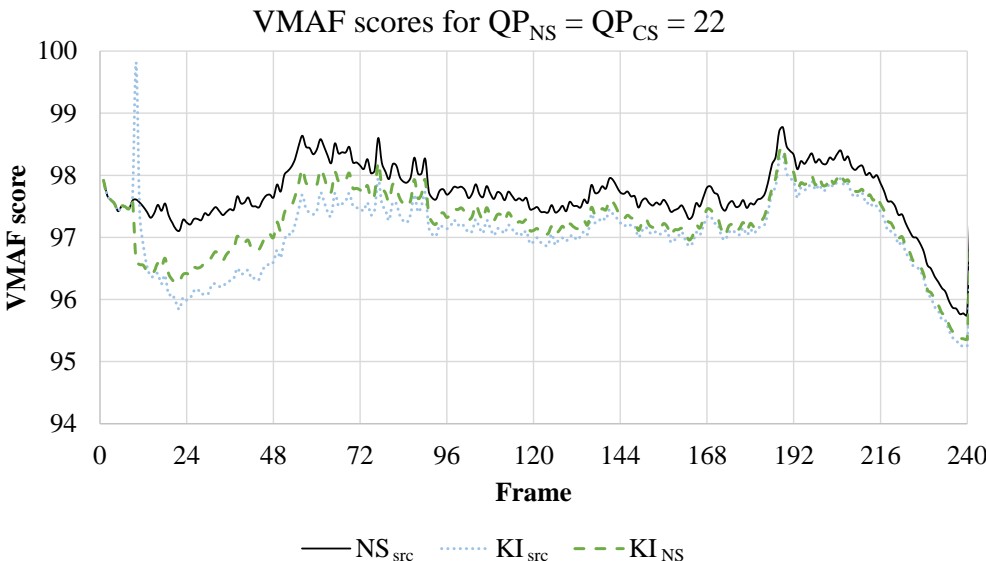

**Figure 4.** Video Multimethod Assessment Fusion (VMAF) scores of the *ParkScene*-sequence, encoded with $QP_{NS} = QP_{CS} = 22$. On average, $KI_{NS}$ has a smaller quality decrease than $KI_{src}$.

To additionally analyze the effect of keyframe insertion on a video with lower quality in depth, Figure 5 shows the VMAF scores for the same sequence as Figure 4, but encoded with $QP_{NS} = QP_{CS} = 32$. Because of the higher QP, the effects that were observed in Figure 4 are more pronounced. That is, the initial quality increase of $KI_{src}$ is much larger: the VMAF score increases with approximately 6 points. Then, until frame #28, the quality remains higher than $NS_{src}$ and $KI_{NS}$. Afterward, the quality from $KI_{src}$ drops below the quality of $KI_{NS}$. On average, and over all frames (excluding the first 9 frames), the VMAF score decrease of $KI_{src}$ is 1.1, whereas it is only 0.6 for $KI_{NS}$. In other words, the quality decrease of $KI_{NS}$ is smaller on average, yet the first frames after keyframe insertion of $KI_{src}$ are of higher quality.

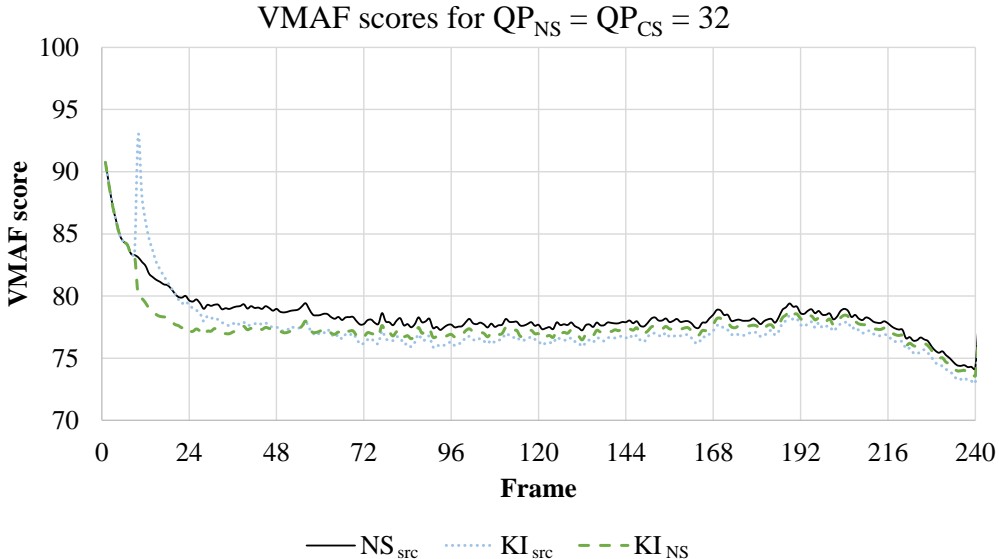

**Figure 5.** VMAF scores of the *ParkScene*-sequence, encoded with $QP_{NS} = QP_{CS} = 32$. It takes several frames for the quality of $KI_{src}$ to drop below the quality of $NS_{src}$ and $KI_{NS}$.

To summarize the impact on the quality for multiple sequences and QP values, Table 2 shows the decrease in VMAF score averaged over all frames and over all tested sequences,

for both $KI_{src}$ and $KI_{NS}$, for all $QP_{NS}$, and for all $QP_{CS}$. An initial observation that can be made is that, for relatively low $QP_{NS}$ and $QP_{CS}$ values, the average VMAF decrease is lower for $KI_{NS}$ than for $KI_{src}$. This observation was also made in Figures 4 and 5. Since the frame size increase of $KI_{NS}$ is smaller than the increase in $KI_{src}$, $KI_{NS}$ is the best performing mode for relatively low $QP_{NS}$ and $QP_{CS}$ values, assuming many frames are streamed before a new keyframe is introduced.

In contrast, when using large $QP_{NS}$ and $QP_{CS}$ values, the quality decrease is slightly larger for $KI_{NS}$ than for $KI_{src}$. The most extreme difference happens for $QP_{NS} = 37$ and $QP_{CS} \leq 32$. In that case, the quality of $KI_{src}$ even increases rather than decreases. That is because the inserted keyframe is of a much higher quality than $NS_{src}$, and therefore has a slightly positive impact on the quality. However, as seen in Table 1, the frame size increase of the inserted keyframe in those cases is between 50.7 and 149.4. Therefore, one can argue that the quality increase is not worth the extra bitrate overhead in those cases. For the other cases where the quality decrease of $KI_{src}$ is smaller than the decrease of $KI_{NS}$, such as for $QP_{NS} \leq 27$ and $QP_{CS} = 32$, the corresponding bitrate increase is (slightly) larger as well.

In general, one can observe that a higher $QP_{NS}$ value (with an equal $QP_{CS}$ value) usually results in a larger VMAF score decrease. As those cases also result in a larger bitrate increase, the proposed keyframe insertion method is least effective for high QPs.

Last, using a higher $QP_{CS}$ value (while keeping $QP_{NS}$ constant) decreases the quality more than when using a lower $QP_{CS}$ value, which is as expected.

**Table 2.** Average decrease in VMAF score.

| | | Average VMAF Score Decrease | | | | | | | |
| | | $KI_{src}$ | | | | $KI_{NS}$ | | | |
| **$QP_{CS}$** | $QP_{NS} =$ | **22** | **27** | **32** | **37** | **22** | **27** | **32** | **37** |
|---|---|---|---|---|---|---|---|---|---|
| 22 | | 0.38 | 0.60 | 0.35 | −0.52 | 0.27 | 0.36 | 0.25 | 0.15 |
| 27 | | 0.61 | 0.78 | 0.48 | −0.40 | 0.59 | 0.60 | 0.44 | 0.28 |
| 32 | | 1.35 | 1.41 | 0.90 | −0.08 | 1.40 | 1.46 | 0.75 | 0.48 |
| 37 | | 3.09 | 2.93 | 1.98 | 0.73 | 3.22 | 2.98 | 2.01 | 0.94 |

4.3.2. Temporal Quality Change

From Table 2, we concluded that the quality of $KI_{NS}$ is better than $KI_{src}$ for low QP values, on average. However, from Figures 4 and 5, we observed the opposite in the first frames after keyframe insertion. That is, in the first frames, $KI_{src}$ is of a better quality than $KI_{NS}$. Additionally, we observed that using a higher QP resulted in more frames before the quality of $CS_{src}$ drops below the quality of $CS_{NS}$.

To analyze more thoroughly how long $KI_{src}$ has a better quality than $KI_{NS}$, Table 3 shows the average number of frames (after keyframe insertion) until $KI_{NS}$ has a better VMAF score than $KI_{src}$. In other words, it shows the number of frames before a quality inversion occurs between $KI_{NS}$ and $KI_{src}$.

When using $QP_{NS} = QP_{CS} = 22$, the quality of $KI_{src}$ is only better than $KI_{NS}$ for 6.1 frames, on average over all test sequences. In contrast, when using $QP_{NS} = QP_{CS} = 32$, it is better for 46.4 frames, on average. These results are in line with the observations of Figures 4 and 5. That is, a larger QP induces more frames in which $KI_{src}$ is of a better quality. Additionally, it can be observed that when $QP_{NS} \neq QP_{CS}$, the number of frames is much higher than when $QP_{NS} = QP_{CS}$.

The results from Table 3 can be used to decide whether it is better to use $CS_{src}$ or $CS_{NS}$ at keyframe insertion, in function of the number of remaining frames in the GOP of $NS_{src}$. For example, if keyframe insertion is required for $QP_{NS} = QP_{CS} = 32$, and the next keyframe in $NS_{src}$ will be introduced in 16 frames, it is probably better to use $CS_{src}$ for keyframe insertion. In contrast, if the next keyframe will not be introduced for 500 frames, than it is probably better to use $CS_{NS}$ for keyframe insertion, as the average VMAF decrease is less in that case (see Table 2).

**Table 3.** Average number of frames (after keyframe insertion) where $KI_{src}$ has a better VMAF score than $KI_{NS}$. Thereafter, $KI_{NS}$ has a better VMAF score than $KI_{src}$ for the first time.

| | | **Frames Before Quality Inversion** | | | |
|---|---|---|---|---|---|
| $QP_{CS}$ | $QP_{NS}$ = | **22** | **27** | **32** | **37** |
| 22 | | 6.1 | 40.9 | 121.1 | 158.9 |
| 27 | | 57.3 | 15.8 | 117.7 | 160.7 |
| 32 | | 13.7 | 94.0 | 46.4 | 160.6 |
| 37 | | 51.6 | 44.3 | 118.1 | 143.5 |

### 4.3.3. PSNR Decrease

For completeness, Table 4 shows the average decrease in PSNR. Compared to using VMAF, $KI_{NS}$ performs better than $KI_{src}$ in more cases (e.g., also when $QP_{NS} = QP_{CS} = 37$). Additionally, when increasing the $QP_{NS}$ value (with a constant $QP_{CS}$ value), the quality decrease remains approximately constant. In contrast, using VMAF, the quality decreased more for high QP values. This difference between PSNR and VMAF is explained by the logarithmic scale of PSNR. For example, for $KI_{src}$, both $QP_{NS} = QP_{CS} = 27$ and $QP_{NS} = QP_{CS} = 37$ have an equal decrease in PSNR of 0.32 dB. However, the PSNR of $NS_{src}$ with $QP_{NS} = 27$ is larger than the PSNR of $NS_{src}$ with $QP_{NS} = 37$. Due to the logarithmic scale, a decrease of 0.32 dB from a large initial PSNR is caused by less pixel changes than from a small initial PSNR. Furthermore, no quality increase compared to $NS_{src}$ is observed for $QP_{NS} = 37$ and $QP_{CS} \leq 32$ in terms of PSNR. Last, we can see a that the quality decreases less when using a larger $QP_{CS}$ value (while keeping $QP_{NS}$ constant), which is similar to the observation in Table 2.

**Table 4.** Average decrease in Peak Signal-to-Noise Ratio (PSNR).

| | | **Average PSNR Decrease (dB)** | | | | | | | |
|---|---|---|---|---|---|---|---|---|---|
| $QP_{CS}$ | | **$KI_{src}$** | | | | **$KI_{NS}$** | | | |
| | $QP_{NS}$ = | **22** | **27** | **32** | **37** | **22** | **27** | **32** | **37** |
| 22 | | 0.34 | 0.18 | 0.10 | 0.03 | 0.24 | 0.15 | 0.06 | 0.05 |
| 27 | | 0.61 | 0.32 | 0.13 | 0.06 | 0.58 | 0.22 | 0.12 | 0.05 |
| 32 | | 1.29 | 0.66 | 0.29 | 0.14 | 1.26 | 0.70 | 0.19 | 0.10 |
| 37 | | 2.41 | 1.35 | 0.62 | 0.32 | 2.46 | 1.30 | 0.65 | 0.19 |

### 4.3.4. SSIM Decrease

Table 5 shows the average decrease in SSIM. The SSIM values of $KI_{src}$ are slightly higher than those of $KI_{NS}$, which is in contrast to their relation in the VMAF scores and PSNR values. However, the differences in SSIM are negligible. The other observations from Tables 2 and 4 regarding the QP values are also observed in Table 5.

**Table 5.** Average decrease in Structural SIMilarity (SSIM).

| | | **Average SSIM Decrease** | | | | | | | |
|---|---|---|---|---|---|---|---|---|---|
| $QP_{CS}$ | | **$KI_{src}$** | | | | **$KI_{NS}$** | | | |
| | $QP_{NS}$ = | **22** | **27** | **32** | **37** | **22** | **27** | **32** | **37** |
| 22 | | 0.05 | 0.06 | 0.03 | −0.09 | 0.04 | 0.06 | 0.04 | 0.03 |
| 27 | | 0.09 | 0.10 | 0.05 | −0.07 | 0.10 | 0.09 | 0.08 | 0.06 |
| 32 | | 0.21 | 0.21 | 0.14 | 0.00 | 0.22 | 0.23 | 0.14 | 0.16 |
| 37 | | 0.50 | 0.51 | 0.40 | 0.20 | 0.51 | 0.52 | 0.46 | 0.27 |

### 4.3.5. Worst-Case Quality Decrease

Although the average VMAF, PSNR, and SSIM decreases remain relatively small, it is important to carefully monitor the worst-case quality decrease in a single frame of

a single tested sequence as well. As the keyframe insertion introduces errors that can drift uncontrollably, the worst-case results may expose obtrusive artefacts. For example, Figure 6 shows the VMAF decrease of $KI_{src}$ compared to $NS_{src}$, for all sequences, with $QP_{NS} = QP_{CS} = 22$. Although the VMAF decrease of most tested sequences stabilize between 0 and 1 VMAF point, the *BQSquare*-sequence has a VMAF decrease of up to 4.3 VMAF points. This worst-case sequence is discussed more thoroughly in Section 4.4.

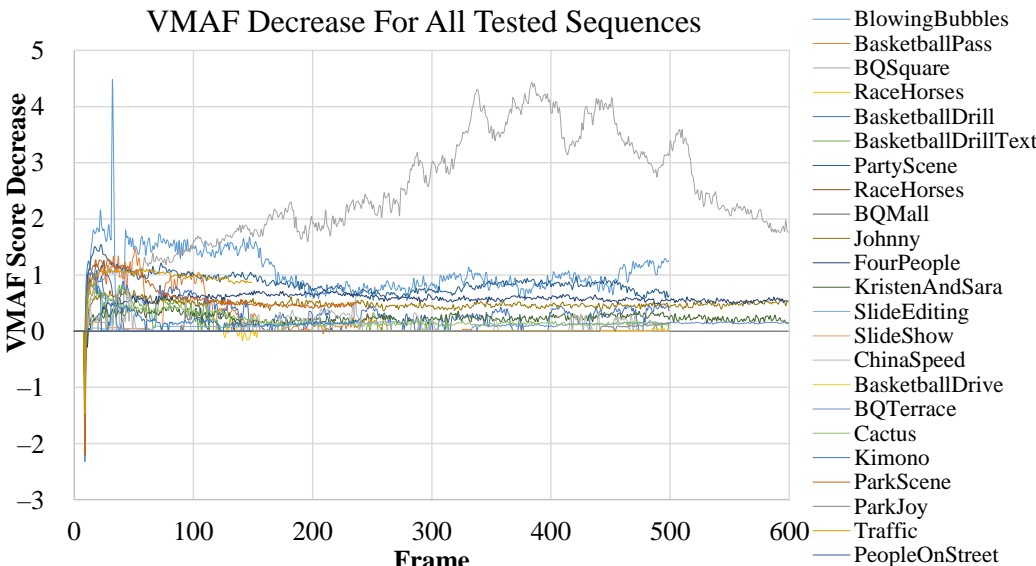

**Figure 6.** VMAF score decreases of $KI_{src}$ of all tested sequence, encoded with $QP_{NS} = QP_{CS} = 22$.

On a higher level, Table 6 shows the worst-case VMAF score decrease for both $KI_{src}$ and $KI_{NS}$, for all $QP_{NS}$, and for all $QP_{CS}$. More specifically, each value in the table was calculated by taking the maximum VMAF decrease over all frames and all tested sequences.

In general, the worst-case VMAF score decrease results are approximately an order of magnitude higher than the average results in Table 2. Additionally, the contrast between $KI_{src}$ and $KI_{NS}$ is even more pronounced. That is, the worst cases of $KI_{src}$ are generally much larger than those of $KI_{NS}$.

**Table 6.** Worst-case decrease in VMAF score.

| | | **Worst-Case VMAF Score Decrease** | | | | | | | |
| $QP_{CS}$ | | $KI_{src}$ | | | | $KI_{NS}$ | | | |
| | $QP_{NS} =$ | **22** | **27** | **32** | **37** | **22** | **27** | **32** | **37** |
| 22 | | **4.45** | 4.97 | 6.60 | 6.48 | **3.08** | 2.96 | 1.22 | 1.41 |
| 27 | | 4.22 | **4.45** | 6.52 | 6.80 | 3.43 | **4.31** | 2.51 | 1.26 |
| 32 | | 6.49 | 6.18 | **7.15** | 6.77 | 6.50 | 7.01 | **3.04** | 2.02 |
| 37 | | 13.40 | 9.03 | 7.32 | **7.52** | 14.39 | 11.01 | 7.83 | **5.17** |

### 4.4. Drift-Error Artifacts Analysis

This section briefly analyzes the artifacts that are introduced due to keyframe insertion, of which the errors drift throughout the GOP.

First, Figure 7c shows the frames corresponding to Figure 4, i.e., from the *ParkScene*-sequence, encoded with $QP_{NS} = QP_{CS} = 22$. More specifically, frame #28 of $NS_{src}$ and $KI_{src}$ is shown, as it corresponds to the largest VMAF score decrease in that sequence, for those QP values (1.29 VMAF points). Additionally, the differences are visualized in Figure 7c, which shows the Y-channel differences using the absolute pixel value difference, and a pixel value is represented by 8 bits. It is quite hard to spot any artifacts in Figure 7b,

even with aid of the visualization of the differences in Figure 7c. In general, most artifacts are in areas with a lot of texture, such as in the leaves of the plants and trees. As the human eye is least sensitive to changes in areas with a high spatial frequency, these artifacts are generally imperceptible.

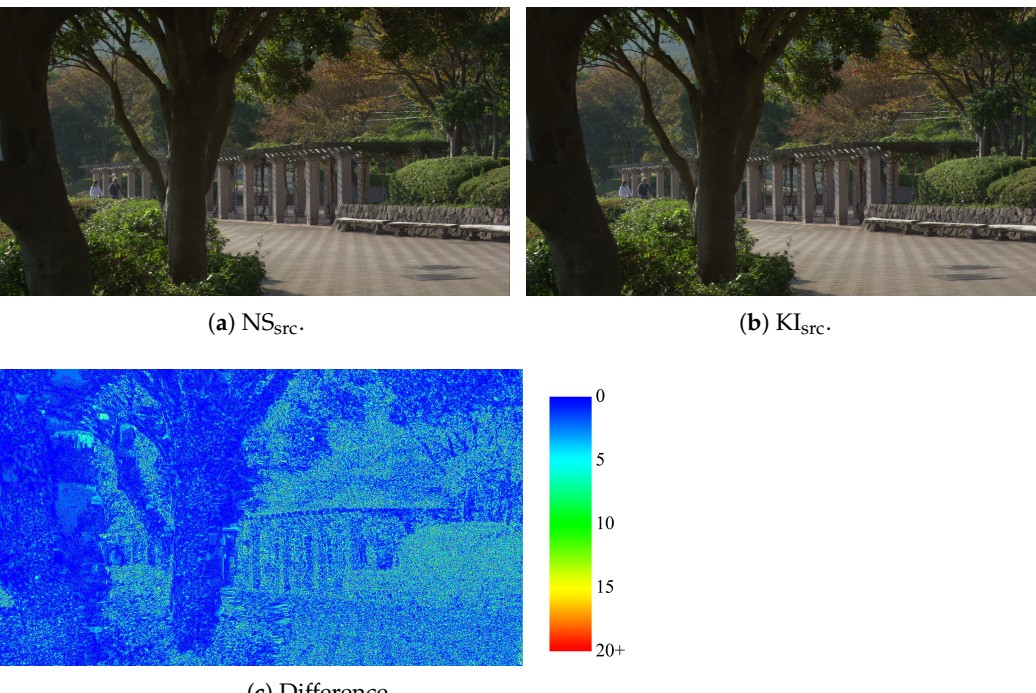

(**a**) $NS_{src}$.

(**b**) $KI_{src}$.

(**c**) Difference.

**Figure 7.** Frames of the *ParkScene*-sequence with $QP_{NS} = QP_{CS} = 27$. (**a**) The frame of $NS_{src}$ without drift-error artifacts. (**b**) The frame of $KI_{src}$ with drift-error artifacts. (**c**) The differences between subfigures (**a**,**b**). It is recommended to view these images in color, digitally.

To more extensively analyze potential drift-error artifacts, one of the worst cases is investigated. Figure 8 shows the frames corresponding to the worst-case VMAF decrease reported in Table 6, of $NS_{src}$ and $KI_{src}$, for $QP_{NS} = QP_{CS} = 27$. More specifically, the VMAF score decrease is 4.45 for the *BQSquare*-sequence at frame #360. At first sight, one may not notice any objectionable artifacts. However, upon close inspection, some local artefacts are visible. For example, color bleeding from the top-right red umbrella into the water can be observed. Moreover, the shadow of the right chair under the top-right umbrella became brighter. Furthermore, an additional shadow is introduced for the right chair of the top-left table without umbrella. Thus, in general, these artifacts are visible upon close inspection but may not even be noticed without reference to the original frame. As a final note, it is important to keep in mind that these artifacts correspond to the worst-case quality decrease. In general, on average, the quality decrease is an order of magnitude smaller, and thus even less perceptible.

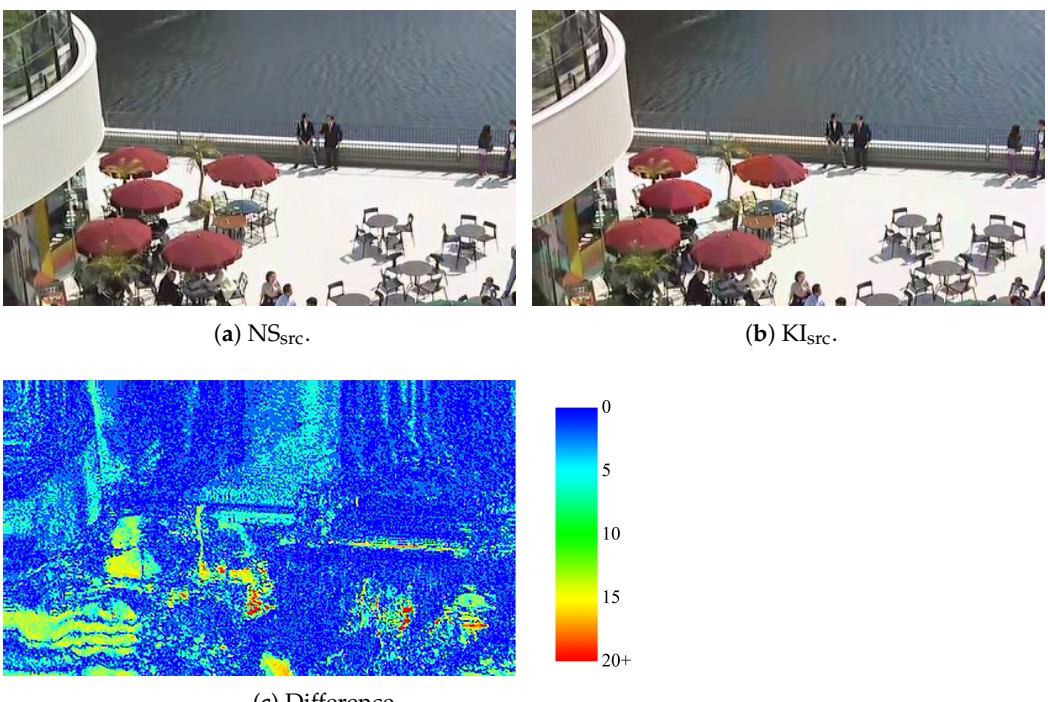

(**a**) NS$_{src}$.

(**b**) KI$_{src}$.

(**c**) Difference.

**Figure 8.** Frames of the *BQSquare*-sequence with QP$_{NS}$ = QP$_{CS}$ = 27, which corresponds to the worst-case VMAF score decrease for those QP values. (**a**) The frame of NS$_{src}$ without drift-error artifacts. (**b**) The frame of KI$_{src}$ with drift-error artifacts. (**c**) The differences between subfigures (**a**,**b**). It is recommended to view these images in color, digitally.

## 5. Example Application

To give a concrete example illustrating the possible workings of the proposed technique, the use case of live Over-The-Top (OTT) video streaming will be described. Imagine a live streaming service where different versions of a stream are generated to accommodate different devices and bandwidth constraints, i.e., the bitrate ladder. A companion stream is generated for every unique resolution present in the bitrate ladder of the live streaming solution. For this specific live stream, it is decided that a maximum random-access latency and packet loss recovery time of one second suffices. Therefore, these companion streams would consist of keyframes every second. Thanks to these companion streams, the keyframe period of the normal stream can be kept minimal, for example, 10 s. All streams will reside on the server until requested by end user devices.

When a device connects, it requests the normal stream and the corresponding companion stream. The companion stream is closed after receiving a keyframe in the normal stream or the companion stream, whichever comes first. In this way, less than a second is needed to join the stream. After joining, less than 10 s of drift artifacts will be experienced corresponding to the measurements presented Table 2. Very specifically, assume that the normal stream (25 fps) and the companion stream are encoded with a QP of 27. Given a 10 s (250 frames) keyframe period in the normal stream, Table 3 recommends to encode the companion stream based on the normal stream, i.e., KI$_{NS}$. This table informs us that on average only the first 15.8 frames have a superior quality when using the source sequence to encode the companion stream. Therefore, it can be safely assumed that after 250 frames in total, the strategy of using the decoded normal stream is most optimal. With this information, Table 2 informs us that a 0.6 decrease in VMAF score will be experienced during less than 10 s. From all the tested sequences, the worst-case reduction in quality during these first seconds was 4.31 VMAF points.

When experiencing a packet loss, the end user device performs regular error concealment and immediately joins the companion stream. Again, the companion stream is closed

after receiving a keyframe in the normal stream or the companion stream, whichever comes first.

Finally, it has to be taken into account that in the normal stream there is only one keyframe every 10 s rather than every second. For the provided settings, Table 1 provides an average factor of 13.9 size increase of a keyframe compared to a predicted frame. A rough estimate based on these numbers tells us that end user devices with reliable connections can enjoy a bitrate reduction in the range of 69% compared to classical streams with a keyframe period of one second and no companion stream configuration.

$$\frac{(13.9 + 249)}{(10 \times 13.9 + 240)} = 0.69$$

In the most extreme example for this application, this also implies that if an end user device would cope with at least one packet loss every second ($\frac{1}{25 fps} = 4\%$ or more loss rate), the bitrate of the stream for this end-user device would have to increase with 45%, i.e., $\frac{1}{0.69}$, compared to the normal stream alone. With the proposed system, this bitrate increase for individual users linearly reduces with a reduction in loss rate. When alternatively tuning the system to the worst performers and giving everyone a stream with keyframe period of a second, everyone would suffer the 45% bitrate increase.

When further comparing to a system where every stream has a keyframe every second to accommodate random access and packet losses, the main downside of the proposed technique is the encoding step to generate the companion stream.

## 6. Discussion

This paper applied the untested concept of keyframe insertion to the H.265/HEVC standard. Additionally, it extensively analyzed the impact on bitrate and quality to evaluate the technique's strengths and limitations, and it evaluated which configurations have the best performance.

This is the first work to thoroughly demonstrate that inserting a keyframe from a companion stream into a normal stream has a limited effect on the bitrate and quality. For example, in our experiments, when using the same QP in the normal and companion stream, the inserted keyframe is approximately 7 to 30 times larger than the inter-frame that it replaces. Thereafter, all remaining frames are received as normal. Contrary to our intuition, the introduced errors generally do not drift uncontrollably and only result in a slight decrease in overall quality. The highest VMAF score differences stay well below a 6-point difference for practical QP values, which is claimed to be the just noticeable [34]. On average, the quality decrease is much smaller, and hence less perceptible. Although these low VMAF scores are frame averages and there may be local perceptible artifacts in some worst cases, these artifacts are not annoying, in our opinion.

In general, the impact on the frame size and quality is lowest when inserting a keyframe from $CS_{NS}$, i.e., when encoding the keyframes of the companion stream from the reconstructed normal stream. Additionally, using low QP values, i.e., a high-quality companion stream, results in a better performance than when using high QP values. Only when relatively few frames are remaining in the GOP of the normal stream, or when using very high QPs, it may be more beneficial to insert a keyframe of $CS_{src}$ rather than a keyframe of $CS_{NS}$.

Using the proposed method and the experimental results, design decisions about codec settings in the H.265/HEVC standard can be made more straightforward. That is, the normal stream takes the responsibility of compression efficiency while the companion stream covers random access and error resilience. As such, the service of high-performing end users does not suffer from the few low-performing devices or users that switch channels. Definitely in the context of low-latency video distribution over error-prone networks, this functionality is very beneficial.

Future work should address the usage of TMVP. Using TMVP in combination with keyframe insertion results in explicit blocking artifacts, and thus has been disabled in

the proposed method. A solution could be to additionally encode an inter-frame after every keyframe in the companion stream, which has the same motion vectors as the corresponding frame in the normal stream.

Furthermore, the proposed work could be extended by introducing predicted frames in the companion stream referencing keyframes from the normal stream. In this way, packet loss problems could be resolved with more efficient predicted frames rather than less efficient keyframes. It must be noted that such technique could only resolve packet loss errors and not accommodate random access.

Finally, future work may investigate the effect on other codecs, such as the older H.264/AVC and the recently standardized H.266/VVC. It would be interesting to know the influence of the codec on the applicability of keyframe insertion.

**Author Contributions:** Conceptualization, G.V.W., J.V. and P.L.; Formal analysis, G.V.W. and H.M.; Funding acquisition, J.V. and P.L.; Investigation, G.V.W. and H.M.; Methodology, G.V.W., H.M. and P.L.; Project administration, P.L.; Software, G.V.W. and H.M.; Supervision, P.L.; Validation, G.V.W. and H.M.; Writing—original draft, G.V.W. and H.M.; Writing—review & editing, J.V. and P.L. All authors took part in the discussion of the work described in this paper. All authors have read and agreed to the published version of the manuscript.

**Funding:** This work was funded in part by the Research Foundation—Flanders (FWO) under Grant 1S55218N, in part by IDLab (Ghent University—imec), in part by Flanders Innovation & Entrepreneurship (VLAIO) project LIVE-G (Low-latency video over five-G—HBC.2020.2375), and in part by the European Union.

**Data Availability Statement:** The results presented in this study are available on http://media.idlab.ugent.be/2021/02/11/keyframe-insertion-enabling-low-latency-random-access-and-packet-loss-repair/ (accessed on 20 March 2021) .

**Acknowledgments:** The computational resources (STEVIN Supercomputer Infrastructure) and services used in this work were kindly provided by Ghent University, the Flemish Supercomputer Center (VSC), the Hercules Foundation, and the Flemish Government department EWI. The distortion maps in Figures 7 and 8 were based on YUVToolkit's visualizations.

**Conflicts of Interest:** The authors declare no conflict of interest. The funders had no role in the design of the study; in the collection, analyses, or interpretation of data; in the writing of the manuscript; or in the decision to publish the results.

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
