# Peer review of "Keyframe Insertion: Enabling Low-Latency Random Access and Packet Loss Repair"

_electronics, doi:10.3390/electronics10060748_

Round 1

Reviewer 1 Report

The paper is dedicated to an efficient key-frames insertion into H.265/HEVC stream in order to provide low-latency random access and packet loss repair. An insertion of a companion stream consisting of key frames are proposed. Simulation results are provided.

Comments to the authors:

1) Since the paper is related to the content based methods, I think the authors should provide comparison with other content based approaches: intra refresh and spatial scalability. In the first case we can pseudo-randomly insert intra blocks at different locations within P-frames in order to guarantee that normal playback would be possible after decoding of N frames, where N could be selected depending on random access requirements or target packet loss rate. In this approach we need to spend a bit more bit for intra blocks, but we don’t need to disable TMVP as in the author approach. Second, we can compress a video using spatial scalability as it is introduced in H.265/HEVC, i.e., we can form the base stream as a set of I-frames with lower resolution. Please discuss these two approaches in Introduction and provide some comparison results if possible.

2) As I see, only constant QP mode is considered. I think the authors should discuss how we need to modify the proposed approach for constant bit rate case.

3) If I understand correctly, the proposed approach assumes some kind of optimization, i.e., QP selection, Intra-frame period for the companion stream and so on. But I cannot see the optimization task as well as its solution. Let say, the basic case is encoding the normal stream only using TMVP, which takes bit rate Cbase (using quantization parameter Qb). In our task we need to disable TMVP and get Cnormal (using quantization parameter Qn) and insert the companion stream with bit rate Ccomp (using quantization parameter Qc). For example, our task is to find the best Q*n and Q*c  for the proposed scheme keeping Cnormal + Ccomp = Cbase (for fair comparison) for a given playback delay. As I result, we also could show the quality decreasing caused by decrease of the playback quality.

4) The authors should explain better how the scheme works under packet losses and how to optimize the parameters (QP and I-frame period for the companion stream) for a given packet loss rate similar to 3).

Author Response

Dear reviewer,

We appreciate the time you took to read, analyze and comment the work we provided you. Therefore, with equal thoroughness, we would like to provide you with all the information we obtained, provided the limited time we had to prepare this.

First, we would like to better describe a possible scenario we envision this work to be used in, because we think there is a misconception about how this technique could be used.

Imagine a live streaming use case where a server produces two versions of the video stream (normal stream and companion stream). We think a misconception happened at this point, because in the application we foresee, the end user hardly ever receives the companion stream. The companion stream resides on the server and a single frame from this companion stream will only be transmitted when an end user requests a frame from it. As a consequence, the end user connection will only consume bitrate necessary for the normal stream. Only when a channel switches to the normal stream, one frame of the companion stream will be provided to start the decoding process of the normal stream. Equivalently, when packets get lost, the end user will request one companion stream frame to startup the normal stream again with less artefacts caused by the packet loss. Consequently, there is hardly any bitrate overhead associated with the proposed technique except on the server.

1) With respect to the first question you wonder if a comparison with intra refresh could be included. There are two ways I could see an intra refresh strategy as a comparison with the proposed technique. Intra refresh could be used in the normal stream without considering any form of companion stream and intra refresh could be used in the companion stream. Let us consider the following comparison:

  • One video stream at QP27 25fps using intra refresh every 2 seconds (50 frames). Intra refresh as a technology approximately competes with intra images because there is no overhead associated with intra refresh. There is an equal amount of intra blocks, but there is an additional restriction of large motion vectors not being allowed to cross the intra refresh frontier. But less us assume this overhead is non existing. So the bitrate of an intra refresh at 2 seconds is equal to an intra frame every 2 seconds. I hope we can agree on that.
  • If we envision superior random access capabilities, our proposed technique in this case would be constructed around an intra period of 10 seconds (250 frames) in the normal stream (let us take QP27) and an intra image at every frame in the companion stream (for fairness also QP27).

Based on table 1, a comparison of these two scenarios could be calculated as follows:
In steady state, when there is no packet loss or no channel switching, bitrate reduction of the normal stream compared to the intra refresh stream can be calculated as follows:

Every 10 seconds (250 frames), the normal stream has 4 less intra frames. With a frame size increase for intra at QP27 of 13.9 this gives us the following relative bitrates:

Intra refresh: 5 x (49 P frames of relative size 1) + (1 I frame of relative size 13.9) = 314.5

Normal stream: (249 P frames of relative size 1) + (1 I frame of relative size 13.9) = 262.9

So, during steady state, using the suggested intra refresh approach, people would be downloading 19.6% more bitrate to achieve the inferior channel switching capabilities.

When there is not a steady state, for example whenever there is a channel switching event, there are different actions that can be performed in the different cases. We described this comparison in section 4.2 Impact on Frame Size and such a comparison can be considered a qualitative comparison with the Client-Based Methods from the state-of-the-art:

“Also, note that alternative solutions either have a high latency or a higher bitrate overhead. First, one can wait for the next keyframe to resume playback, which may be a very long time (e.g. up to 600 frames). Not only does this have a high latency, it also wastes bandwidth potential during the waiting period.”

In the provided example, this would give an average wait time of 1 second for the intra refresh scenario compared to no wait time for the propose technique.

“Second, an alternative solution for fast channel switching is to request the keyframe from the current segment at the new channel (from the normal stream), as well as all inter-frames up until the current frame. Then, these are all decoded and optionally displayed on-screen with a fast playback speed. This results in a high latency and requires more bandwidth than the proposed keyframe insertion (which requests only a single keyframe to start playback of the current frame at the new channel).”

When you consider intra refresh in the normal stream accompanied with additional information to complete the intra refresh to full intra frames, this is what was proposed in the referenced paper: Jennehag, U.; Döhla, S.; Fuchs, H.; Thoma, H.  Gradual tune-in pictures for fast channel63change.  2011 IEEE Consumer Communications and Networking Conference (CCNC), 2011,64pp. 766–770.
With respect to bitrate and quality, similar results to the proposed technique can be assumed based on the premise that intra refresh provides no overhead to the stream.

Using spatial scalability in HEVC would be very inefficient when comparing with the proposed technique.

As mentioned in the state-of-the-art: “Most works propose to provide the low-quality companion stream as a base layer enhanced with the normal stream as enhancement layer [27]. During steady state watching, these techniques use both the base and enhancement layers of the selected channel to achieve full quality.  Fast channel switching and error recovery is then performed by decoding only the base layer containing numerous keyframes. The downside of this technique is the continuous presence of a low-efficiency base layer which causes an overhead associated with scalable coding.”

As mentioned at the beginning of this reply, the proposed technique never provides the companion stream to the end user. Only specific frames are requested when needed. With spatial scalable coding, the inefficient baselayer containing all the I frames would be transmitted all the time to the end user, because the enhancement layer needs this to decode. In my view, comparing with spatial scalability would be unfair. The only technique that could compete is single loop scalable coding with the companion stream encoded as enhancement layer (see [28]). But this technology did not make it inside HEVC which turns it irrelevant to compare with I believe.

Finally, with respect to the first question, we would like to emphasize the impact of disabling TMVP. Following your comment, we did the measurements and the following results have been obtained:

Average: -2.57 BD-rate
Std: 0.86 BD-rate
Min: -3.92 BD-rate
Max: -0.22 BD-rate

So 2.57% bitrate decrease can be expected by enabling TMVP. Although this reduction is not insignificant, especially in lossy scenarios such an overhead of disabling TMVP is still advised to be made. When you enable TMVP, you should provide not only error concealment for pixels, but also for motion vectors. When the decoder makes mistakes in concealing pixel information, the impact is small and unnoticeable. Performing error concealment on motion vectors has an enormous negative impact. Therefore, in the proposed use cases, TMVP will be already disabled in a lot of cases.
The numbers and context information have been added to the paper.

2) In the second question, you inform about the impact on CBR. With constant bitrate, the quality influence of bitrate variation can impact the results a lot and therefore was left untouched. For constant bitrate, there are startup issues at the beginning of the video where the rate control algorithm has no statistics yet about the behavior of the video. This definitely has a significant impact on small test sequences such as the ones used in this paper and elsewhere. Therefore, you will see minimal research findings documenting CBR. Additionally, with CBR, you need to specify a buffer size, namely the VBV buffer. CBR with an infinite buffer size is VBR as has been tested. When setting CBR with a VBV buffer of a single frame, every frame is of equal size.

What can already be done with the existing measurements is getting an idea of the quality impact when more or less requiring the intra frame to be of a similar bitrate as the entire sequence. To do this, the QP needs to be increased in table 1 to match constant bitrate.

For example, let us assume a QP of 22 and we would like to have the companion stream to have the same bitrate as the normal stream. The closest we can get is an Intra frame with QP 37 which is still twice the size of the P frames in the QP22 sequence.  Taking a QP of the normal stream at 22 and the QP of the companion stream at 37 will give you VMAF reductions of 3.22 in table 2. This is among the worst case scenarios that we tested. But this is compared to VBR of course. In case of a fair comparison with a similar CBR, similar results as the ones reported can be expected. I do not see any reason why the proposed technique would be penalized with a smaller VBV buffer size.

3) In the 3rd question, you ask about optimization, i.e., QP selection, Intra-frame period for the companion stream. Actually, this is a design decision, not an optimization. Therefore, we provided as much data as we could such that the trade-off can be made based on requirements. Let me give an example. The best strategy is to have a companion stream with an intra frame for every frame. The downside of doing this is processing, but the processing cost is not ours to trade-off. If you have an application that allows a large processing budget, there is not trade-off to optimize. Just encode every frame in the companion stream. The size of the companion stream depends on the latency that you choose. Again, when designing a system, the use case requires a certain latency and thus a certain buffering time. It is for the application developer to decide on the buffering time and to choose the frame size increase of the I frame accordingly. But this restriction has to be made for the normal video stream as well. So, the suggestion is to set the companion stream I frames of a size equivalent to the I frame size of the normal stream. This is a choice of the one designing the system and using the proposed paper, these designers can now analyze what the quality impact will be when choosing the I frame size.

In your question, you propose to compare Cbase with the sum of Cnormal and Ccomp. As explained in the beginning, such comparison would be unfair because the normal stream and the companion stream are never transmitted at the same time. There is no reason to add up the bitrates of the normal stream and the companion stream because these streams are only present on the server, not on the network or at the client.

4) Again, no optimization takes place here. If there is packet loss and there is processing capacity, the best thing to do is generate an I frame in the companion stream for every frame. Then the latency needs to be chosen as is always done. This latency defines the buffers and the buffers define the I frame size. The I frame size can be related to a QP difference in Table 1 and the quality impact can be derived in table 2.

Reviewer 2 Report

The paper proposes a key frame insertion method to address the problem of random access and packet loss. 

The idea is very easy to understand and the authors provide an extensive evaluation results. 

One concern is that the evaluation is only among their own proposed methods, not with other existing methods. 

As the authors mention in section 2, there are many existing works, especially the content based methods. 

Thus, it would have been more helpful if they had compared their methods with some of the existing methods. 

Another concern is about the experiment method. 

In their description, it is not clear whether they artificially drop the P frames at f=9 or they just play the video at arbitrary time. 

It would have been more helpful if they had described their experiment method more clearly. 

Author Response

Dear reviewer,

We appreciate the time you took to read, analyze and comment the work we provided you. Therefore, with equal thoroughness, we would like to provide you with all the information we obtained, provided the limited time we had to prepare this.

First, we would like to better describe a possible scenario we envision this work to be used in, because we think there is a misconception about how this technique could be used.

Imagine a live streaming use case where a server produces two versions of the video stream (normal stream and companion stream). We think a misconception happened at this point, because in the application we foresee, the end user hardly ever receives the companion stream. The companion stream resides on the server and a single frame from this companion stream will only be transmitted when an end user requests a frame from it. As a consequence, the end user connection will only consume bitrate necessary for the normal stream. Only when a channel switches to the normal stream, one frame of the companion stream will be provided to start the decoding process of the normal stream. Equivalently, when packets get lost, the end user will request one companion stream frame to startup the normal stream again with less artefacts caused by the packet loss. Consequently, there is hardly any bitrate overhead associated with the proposed technique except on the server.

1) In the first question, you raise the concern of not having a comparison of the proposed technique with the state-of-the-art. We would like to confirm that indeed, we did not provide any quantitative comparison but rather a qualitative comparison. Such comparison is present in the state-of-the-art section and the result section:

Comparing with SI/SP pictures would be interesting but purely academic because HEVC does not support any form of switching pictures. The closest technology to switching pictures is what we propose. This is mentioned in the paper as follows :” Additionally, SI/SP picture coding has not been widely adopted in H.264/AVC encoders and decoders and the compression tool has not been included in H.265/HEVC.”

Techniques like [4] and [5] are based on the same general idea but applied to older standards. A direct comparison would rather provide with details about differences in standards and would therefore be irrelevant.

Using spatial scalability in HEVC would be very inefficient when comparing with the proposed technique. As mentioned in the state-of-the-art: “Most works propose to provide the low-quality companion stream as a base layer enhanced with the normal stream as enhancement layer [27]. During steady state watching, these techniques use both the base and enhancement layers of the selected channel to achieve full quality.  Fast channel switching and error recovery is then performed by decoding only the base layer containing numerous keyframes. The downside of this technique is the continuous presence of a low-efficiency base layer which causes an overhead associated with scalable coding.”

As mentioned at the beginning of this reply, the proposed technique never provides the companion stream to the end user. Only specific frames are requested when needed. With spatial scalable coding, the inefficient baselayer containing all the I frames would be transmitted all the time to the end user, because the enhancement layer needs this to decode. In my view, comparing with spatial scalability would be unfair. The only technique that could compete is single loop scalable coding with the companion stream encoded as enhancement layer (see [28]). But this technology did not make it inside HEVC which turns it irrelevant to compare with I believe.

So, comparing with content-based techniques would not be fair, because these could not be applied in practice.

In the result section, we qualitatively compare with different client-based methods.

When there is not a steady state, for example whenever there is a channel switching event, there are different actions that can be performed in the different cases. This is described in section 4.2 Impact on Frame Size as follows:

“Also, note that alternative solutions either have a high latency or a higher bitrate overhead. First, one can wait for the next keyframe to resume playback, which may be a very long time (e.g. up to 600 frames). Not only does this have a high latency, it also wastes bandwidth potential during the waiting period.”

We could perform measurements to prove this, but that would not make any sense. For example, if we take

  • One video stream at 25fps using intra frames every 2 seconds (50 frames).
  • If we envision superior random access capabilities, our proposed technique in this case would be constructed around an intra period of 10 seconds (250 frames) in the normal stream and an intra image at every frame in the companion stream.

Measuring such setup would provide a channel switching latency of 1 frame for the proposed technique compared to an average of 25 frames latency for the other technique. The bandwidth difference would not make sense because there is no bandwidth overhead if you are otherwise just waiting for an I frame. You have bandwidth to display something useful but not using it. So performing such actual measurement did not seem useful for us.

“Second, an alternative solution for fast channel switching is to request the keyframe from the current segment at the new channel (from the normal stream), as well as all inter-frames up until the current frame. Then, these are all decoded and optionally displayed on-screen with a fast playback speed. This results in a high latency and requires more bandwidth than the proposed keyframe insertion (which requests only a single keyframe to start playback of the current frame at the new channel).”

Also here, we could perform a measurement, but conceptually it is already clear what the gains would be.

2) A concern was provided with respect to the experiment method. We believe there was some confusion about what operation we performed on the 9th frame. To resolve this concern we have rewritten the text dealing with the experiment method of frame 9.

“In the experiments, we inserted a keyframe at frame f=9, which is the 10th frame and started decoding from there on as if a random access was performed or as if frame f=8 had been lost. This is an arbitrary decision and could have been any other frame in the beginning of the sequence. Given the set of sequences we used, the 10th frame is taken to avoid duplicated frames and black frames present in the first frames of some sequences. These duplicate and black frames distorted the results heavily because their rate would be very small”

Reviewer 3 Report

The manuscript discussed the keyframe Insertion to enabling low-latency random access and packet loss repair. 

The authors have presented the work in a very elaborate manner. The results are also presented with good clarity.

I would like to recommend to accept the manuscript in present form. 

Author Response

Dear reviewer,

We would like to express our appreciation towards the reviewer for reading and validating our work.

Kind regards,

Round 2

Reviewer 1 Report

I think, the paper still needs to be significantly improved, since it is unclear for the reviewer, i.e., it will be highly likely unclear for a reader as well.

1)” The companion stream resides on the server and a single frame from this companion stream will only be transmitted when an end user requests a frame from it. As a consequence, the end user connection will only consume bitrate necessary for the normal stream.” First of all, I think the authors should show it clearly in the paper. Second, I was confused by this answer since the authors claims “Low-Latency Random Access and Packet Loss Repair”. Since the companion stream is transmitted only in on-demand mode, it means that the considered solution is not low-latency, it is long-latency, since the time needed for the bit stream access and/or packet loss repair depends on a network delay. In video conferencing we cannot resend a frame on demand since such solution can stop playback at the receiver, in video broadcasting we cannot resend a frame for each user, since the resending bit rate for many users could be much higher than the network capacity (imagine transmission of one stream to 10000 users with different channel conditions). So, I think the authors should very clearly describe all set of applications where the proposed approach can be used. If the set of applications is something like video on demand, where playback latency is not crucial, then in case of losses we can resend the same P or B frame. Why we need to send I frame if we can send the same lost P frame? I think the authors should explain all of these issues carefully.

2) “… using the suggested intra refresh approach, people would be downloading 19.6% more bitrate to achieve the inferior channel switching capabilities.” Even if these calculations holds, they are valid only when there are no packet losses. For example, if we have packet loss rate around 1% (typical for video conferences and other wireless networks), then we could have a situation when many P frames between I frames are broken, i.e., the proposed approach will send many key frames to recover the playback which will overload the network.  So, the authors should clearly show the values of the packet loss rates in which the proposed solution works.

3) “CBR with an infinite buffer size is VBR as has been tested”. I’m wondering why the authors assumes infinite buffer size considering video transmission. It is very unrealistic, since in practice the buffer is limited. “When setting CBR with a VBV buffer of a single frame, every frame is of equal size.” is not correct, since rate control chooses different frame sizes for different frames depending on frame complexity and frame type (I or P, B).

Author Response

Dear reviewer,

I thank you for your patience and help in clarifying the proposed solution as I see there is still a misunderstanding.

1) We deliberately did not want to narrow down the research to a specific application because we think it is more broadly applicable than a single use case. We also understand that without a specific use case, it is difficult to follow the entire procedure. Therefore, we hope that the addition of a section named “Example application” resolves your concerns. For your ease, we included the section here:

--------------------begin new section ----------------------------------------

Example application

To give a concrete example illustrating the possible workings of the proposed technique, the use case of live Over-The-Top (OTT) video streaming will be described. Imagine a live streaming service where different versions of a stream are generated to accommodate different devices and bandwidth constraints, i.e. the bitrate ladder. A companion stream is generated for every unique resolution present in the bitrate ladder of the live streaming solution.  For this specific live stream, it is decided that a maximum random access latency and packet loss recovery time of one second suffices. Therefore, these companion streams would consist of keyframes every second. Thanks to these companion streams, the keyframe period of the normal stream can be kept minimal, for example 10 seconds. All streams will reside on the server until requested by end-user devices.

When a device connects, it requests the normal stream and the corresponding companion stream. The companion stream is closed after receiving a keyframe in the normal stream or the companion stream, whichever comes first. In this way, less than a second is needed to join the stream. After joining, less than 10 seconds of drift artefacts will be experienced corresponding to the measurements presented Table 2. Very specifically, assume that the normal stream (25 fps) and the companion stream are encoded with a QP of 27. Given a 10 seconds (250 frames) keyframe period in the normal stream, Table 3 recommends to encode the companion stream based on the normal stream, i.e. KI_NS. This table informs us that on average only the first 15.8 frames have a superior quality when using the source sequence to encode the companion stream. Therefore, it can be safely assumed that after 250 frames in total, the strategy of using the decoded normal stream is most optimal. With this information, table 2 informs us that a 0.6 decrease in VMAF score will be experienced during less than 10 seconds. From all the tested sequences, the worst-case reduction in quality during these first seconds was 4.31 VMAF points.

When experiencing a packet loss, the end-user device performs regular error concealment and immediately joins the companion stream. Again, the companion stream is closed after receiving a keyframe in the normal stream or the companion stream, whichever comes first.

Finally, it has to be taken into account that in the normal stream there is only one keyframe every 10 seconds rather than every second. For the provided settings, Table 1 provides an average factor of 13.9 size increase of a keyframe compared to a predicted frame. A rough estimate based on these numbers tells us that end-user devices with reliable connections can enjoy a bitrate reduction in the range of 69\% compared to classical streams with a keyframe period of one second and no companion stream configuration.

(13.9+249)/(10*13.9+240) = 0.69

This also implies that if an end-user device would cope with one packet loss every second (4% loss rate), the bitrate of the stream for this end-user device would have to increase with 45%, i.e. 1/0.69, compared to the normal stream alone. With the proposed system, this bitrate increase for individual users linearly reduces with a reduction in loss rate. When alternatively tuning the system to the worst performers and giving everyone a stream with keyframe period of a second, everyone would suffer the 45% bitrate increase.

When further comparing to a system where every stream has a keyframe every second to accommodate random access and packet losses, the main downside of the proposed technique is the encoding step to generate the companion stream.

--------------------end new section ----------------------------------------

To be clear, there is no retransmission or stalling of the video in any way. The paper described it as follows:” Whenever a random-access event or packet-loss event occurs, the following procedure needs to be followed (see Figure 2). After a switch or loss event, let us denote the Picture Order Count (POC) of the first occurrence of a keyframe in the companion stream as f. At that moment, the keyframe at position f will be used for starting the decoding process instead of the predicted frame at POC=f.  After that, the regular decoding process can continue with decoding f+1.”

Based on your feedback, we added the following sentence to this paragraph:” With packet loss, this procedure implies that the regular decoding process continues from the moment packets are lost until the first keyframe in the normal stream or the companion stream, whichever comes first. Only then at POC=f, the keyframe is decoded and regular decoding of the normal stream continues from POC=f+1.”

2) We hope that with the addition of the section on example application, the concern of packet loss is covered. We included a specific setting with a worst case 4% packet loss example, but we would like to stress that this paper tries to provide the tools to let streaming architects design a system whatever the properties are. If the architect indeed decides that for its solution there is too much packet loss to handle a large amount of individual requests for companion frames, then the system should adapt to a scenario where more keyframes are included in the normal stream.

3) The buffer mentioned in our answer is the video buffering verifier (VBV) buffer, not the jitter/network buffer. This buffer does not determine the delay or latency in the system. This is a buffer defined by codec engineers to quantify the variation in frame size between different frames. The VBV buffer is monitored by the rate control algorithm. The rate control algorithm will be forced to make every frame of the same size when this buffer has the size of one frame. So, the complexity and the frame type will change, but the rate control will have to manage to get the size equal. The VBV buffer defines the behavior of the rate control. How can you otherwise choose how constant your constant bitrate video is? The VBV buffer has thus nothing to do with the delay in the network. This is where the difficulty lies when testing with rate control. The network jitter determines the jitter buffer and the network variation and occupation determines the VBV buffer.

So in general, with constant bitrate, the quality influence of bitrate variation can impact the results a lot and therefore was left untouched. For constant bitrate, there are startup issues at the beginning of the video where the rate control algorithm has no statistics yet about the behavior of the video. This definitely has a significant impact on small test sequences such as the ones used in this paper and elsewhere. Therefore, you will see minimal academic research findings documenting CBR.

Round 3

Reviewer 1 Report

1) Please specify why 4% packet loss is the maximum one? Why not 5.5% or 6%?

2) As I mentioned before, I don’t think that sending I frame for each packet loss is efficient. For example, if 4% loss happened, then a maximum 10 frames are corrupted. If we able to send an additional information to a user, why do you propose to send 10 intra frames instead of sending these 4% of lost packets? Or we could send the whole corrupted P frame again - it also takes less bits. Finally, we could compress a single frame from the companion stream as P frame using the last received I frame from the normal stream as a reference. Please clarify, why these approaches are not considered.

Author Response

Dear reviewer,

We would like to thank you for your efforts in providing feedback for our proposed work. We feel that after the different iterations it benefitted the readability and understandability of the paper.

1) Please specify why 4% packet loss is the maximum one? Why not 5.5% or 6%?

Indeed, the packet loss can be higher as well, but with the proposed example of one keyframe per second in the companion stream, the system would not be able to do a better job at correcting the problems. Assuming one Network Abstraction Layer Unit (NALU) per frame and thus a frame drop for every lost part of the frame, one divided by 25fps results in 4% loss. To not limit to 4% loss, but to allow higher losses as well, I changed the wording as follows:

“In the most extreme example for this application, this also implies that if an end-user device would cope with at least one packet loss every second (1/(25fps) = 4% or more loss rate), the bitrate of the stream for this end-user device would have to increase with 45%, i.e. 1/0.69, compared to the normal stream alone.”

2) As I mentioned before, I don’t think that sending I frame for each packet loss is efficient. For example, if 4% loss happened, then a maximum 10 frames are corrupted. If we able to send an additional information to a user, why do you propose to send 10 intra frames instead of sending these 4% of lost packets? Or we could send the whole corrupted P frame again - it also takes less bits.

To come back to the previous review round, retransmission of lost frames is not an option because of low-latency constraints. So, as described in Section 3 on materials and methods (line 202) whenever a loss occurs, the end-user client device is free to request the next keyframe from the companion stream and stop possible drift effects caused by the packet loss. Never in this scenario is a retransmission of a lost frame requested. This aspect has been clarified further in the paper as follows: “With packet loss, this procedure assumes that the regular decoding process continues from the moment packets are lost (implying packet-loss drift errors) until the first keyframe in the normal stream or the companion stream, whichever comes first.”

Finally, we could compress a single frame from the companion stream as P frame using the last received I frame from the normal stream as a reference. Please clarify, why these approaches are not considered.

This is an interesting suggestion, but it could only be applied to packet loss scenarios. For random access purposes, a P frame would not be able to start up the streaming process. For packet losses only, it would indeed be beneficial to construct a predicted frame using received keyframes because it would be more compression efficient than an intra predicted frame.

We included the suggestion in future work as follows:

“Furthermore, the proposed work could be extended by introducing predicted frames in the companion stream referencing keyframes from the normal stream.  In this way, packet loss problems could be resolved with more efficient predicted frames rather than less efficient keyframes.  It must be noted that such technique could only resolve packet loss errors and not accommodate random access.”